# Deep Pools: Ecological Sanctuaries for *Steindachneridion melanodermatum*, a Large Endemic and Endangered Pimelodid of the Iguaçu River



**Lucileine de Assumpção** [1,*][ID], **Maristela Cavicchioli Makrakis** [1][ID], **Jhony Ferry Mendonça da Silva** [1][ID], **Karane Allison Silvestre de Moraes** [1][ID], **Suelen Fernanda Ranucci Pini** [1,2], **Patrícia Sarai da Silva** [1], **Elaine Antoniassi Luiz Kashiwaqui** [1,3], **Eduardo Gentil** [4][ID], **Lenice Souza-Shibatta** [5][ID], **Oscar Akio Shibatta** [5] **and Sergio Makrakis** [1,*][ID]

1. Programa de Pós-Graduação em Recursos Pesqueiros e Engenharia de Pesca, Grupo de Pesquisa em Tecnologia em Ecohidráulica e Conservação de Recursos Pesqueiros e Hídricos—GETECH, Universidade Estadual do Oeste do Paraná—UNIOESTE, Rua da Faculdade, 645, Jardim Santa Maria, Toledo 85903-000, PR, Brazil; maristela.makrakis@unioeste.br (M.C.M.); jhonyferry@hotmail.com (J.F.M.d.S.); karanesilvestre@gmail.com (K.A.S.d.M.); suelen.pini@hotmail.com (S.F.R.P.); saraipatricia@gmail.com (P.S.d.S.); elainealk@uems.br (E.A.L.K.)
2. Instituto Federal de Educação, Ciência e Tecnologia de Mato Grosso do Sul—Campus Coxim Rua Salime Tanure s/n, Bairro Santa Tereza, Coxim 79400-000, MS, Brazil
3. Grupo de Estudos em Ciências Ambientais e Educação—GEAMBE, Universidade Estadual de Mato Grosso do Sul—UEMS, BR 163 Km 20.2, Bairro Universitário, Mundo Novo 79980-000, MS, Brazil
4. Grupo de Gestão, Ecologia e Tecnologia Marinha—GTMar, Universidade do Estado de Santa Catarina—UDESC Campus Laguna, Rua Cel. Fernandes Martins, 270, Progresso, Laguna 88790-000, SC, Brazil; eduardo.gentil@udesc.br
5. Museu de Zoologia, Universidade Estadual de Londrina—UEL, Rodovia Celso Garcia Cid—PR 445, Km 380, Londrina 86051-990, PR, Brazil; lenicesouza@hotmail.com (L.S.-S.); shibatta@uel.br (O.A.S.)
*   Correspondence: lucileinea@hotmail.com (L.d.A.); sergio.makrakis@unioeste.br (S.M.)

**Abstract:** *Steindachneridion melanodermatum*, an endemic and endangered species, is the largest catfish in the Lower Iguaçu River basin. Currently, the wild population of this species no longer exists in most of this basin, probably due to the loss of habitat by successive hydroelectric dams. This study evaluated the spatial distribution and abundance of *S. melanodermatum* in the last free-flowing river stretch of approximately 190 km downstream from the Salto Caxias Dam, upstream of the Iguaçu Falls, as well as some tributaries. Hydroacoustic and fishing campaigns were performed between September 2010 to December 2016 to characterize the preferred habitat. A total of 180 specimens was sampled with higher abundances in a protected area near Iguaçu Falls in the Iguaçu National Park. Twenty-four deep pools were identified with maximum depths ranging from 5 to 25 m: 21 pools were along the main channel of the Iguaçu River, and three pools were in tributaries. Deep pools are preferential habitats for this species, especially the deep pool called Poço Preto (P23) and its surroundings. Conservation strategies are required to protect these habitats and prevent *S. melanodermatum* from becoming extinct, such as establishing deep pools as ecological sanctuaries, intensifying the inspection of illegal fishing, and maintaining dam-free tributaries.

**Keywords:** catfish; threatened species; mapping; habitat

## 1. Introduction

*Steindachneridion melanodermatum* is a large catfish (>150 cm in total length) of the family Pimelodidae and is one of the six endemic and endangered species of the genus [1,2]. Its geographic distribution is restricted to the Lower Iguaçu River basin [3–6], where it is known as surubim do Iguaçu, bocudo, or monjolo. This species is the largest fish in the Iguaçu river basin and is possibly migratory [4,7,8], inhabiting free-flowing river sections with fast waters, rocky bottoms, and high depth [3].

Currently, the wild population of *S. melanodermatum* no longer exists in most of the Lower Iguaçu River basin, probably due to the increase of the anthropogenic pressure derived from the loss of habitats to successive hydroelectric dams in this basin [8,9]. Hydroelectric dam constructions affect several fish species due to changes in the freshwater environment, such as substantial habitat loss, fragmentation, and destruction of riparian forests [10–19].

Habitat changes have exposed South American fish species to rapid and expressive biomass declines and extinction threats [14,20–22]. The conservation status of South American freshwater fish varies widely by region [23] and, specifically for Brazil, 10% of freshwater ichthyofauna is categorized as endangered [24]. The preservation of aquatic habitats is an urgent [25] and great challenge due to the many anthropogenic actions imposed on these environments [14]. Moreover, the emerging threats to freshwater biodiversity require renewed consideration [18]. Conservation strategies are needed to recover declining fish populations and should be developed to address several predatory fishing activities, habitat degradation, and blocking migrations [26,27]. Few efforts have been applied to conservation considering the large number of freshwater fish species [28]. Establishing effective conservation strategies will need urgent knowledge about the abundance, distribution, and required habitat of these fish species, as well as the threats to them caused by human pressure [27,29].

Eighty-four percent (84%) of river areas in Brazil are outside of the protected areas, and the coverage of them in large rivers is relatively low, requiring urgent measures for their preservation [28]. The diversity and spatial distribution of freshwater fish species are still poorly understood, making it difficult to determine and implement protected areas [19,30]. One significant challenge for the conservation of large catfish species is the lack of knowledge and the scarcity of protected areas and species conservation plans for freshwater species [31]. Understanding the distribution and size of fish populations, the habitat characteristics, and the interaction of biotic and abiotic factors within an ecosystem [32] are necessary for the management and conservation of fish communities [33,34], especially those that are endangered [35].

The last refuge of the wild population of *S. melanodermatum* is the 190 km long dam-free stretch of the Iguaçu River and its tributaries, which extend downstream from the Salto Caxias Dam to the Iguaçu Falls [36]. The region exhibits diverse landscape characteristics, from a protected area, namely, the Iguaçu National Park (INP), to anthropized ones [37]. The establishment of management and conservation plans for *S. melanodermatum* is crucial because of the upcoming threats to its conservation [37]. Some fish species, such as *S. melanodermatum*, are more vulnerable to environmental degradation since they require specific habitats [38]. Considering the rarity of the wild population [3], the difficulty of sampling them in the field, and limits on the captured specimens that can be used for biological studies [3,5], there are still many gaps in the knowledge of *S. melanodermatum* [36]. Thus, this study focused on evaluating the spatial distribution and abundance of *S. melanodermatum*. Hydroacoustic and fishing campaigns were performed to characterize the preferred habitat in the last free-flowing river. Three hypotheses were tested: (i) the occurrence of species is associated with deep pool sites in the main channel and tributaries, (ii) sites from protected areas have higher abundances than those from unprotected areas, and (iii) deep pool areas are hotspots or sanctuaries for this species once we supposed that these areas could be present in a higher abundance in comparison to other river sites. Moreover, considering the close relationship between deep pools and this fish species, we described these sites in detail to better understand this species' preferred habitat. These findings will assist in the development of urgent conservation strategies for this large endemic and endangered catfish.

## 2. Materials and Methods

### 2.1. Study Area

The Iguaçu River basin is 72,000 km$^2$ and borders Brazil (79% in the State of Paraná, 19% in the State of Santa Catarina) and Argentina (2%) [39,40]. The Iguaçu River is one of the leading and largest tributaries of the Paraná River basin, running approximately 1200 km from its source in the Serra do Mar to the confluence with the Paraná River [41]. It is subdivided according to the geomorphological characteristics in Upper Iguaçu (1st plateau), Middle Iguaçu (2nd plateau), and Lower Iguaçu (3rd plateau) [6].

The Lower Iguaçu River is characterized by a high drop with rapids and waterfalls, such as Salto Santiago (40 m), Salto Osório (30 m), and Salto Caxias (67 m) [41], which enabled the formation of five hydroelectric reservoirs in sequence in the main channel of the Iguaçu River [4,40,42]. In the Lower Iguaçu River, there are also the great falls, namely, the Iguaçu Falls (72 m), in a protected area found in the Iguaçu National Park bordering with Argentina, which promotes the geographical isolation of the distinct upstream ichthyofauna from the downstream ichthyofauna.

The study area comprised the last free-flowing river stretch of the Lower Iguaçu basin, which is approximately 190 km long from downstream of the Salto Caxias Dam (1240 MW) to upstream of the Iguaçu Falls (Figure 1) and exhibits diverse landscape characteristics [37]. In this stretch, the Iguaçu River is embedded with a rocky bed consisting of magnetic basaltic rock that originated in the Mesozoic and presents the remaining fragments of the Atlantic Forest biome. Between the downstream of the Salto Caxias Dam to the upper limit of the Iguaçu National Park, the region is subject to anthropic actions (livestock, agriculture, dam) where a new hydroelectric power plant is located. The new Baixo Iguaçu Hydroelectric Power Plant (350 MW) is located 30 km downstream from the Salto Caxias Dam, whose dam axis is approximately 0.7 km from the upper limit of Iguaçu National Park [36]. However, the samplings for this study were conducted before this dam's construction.

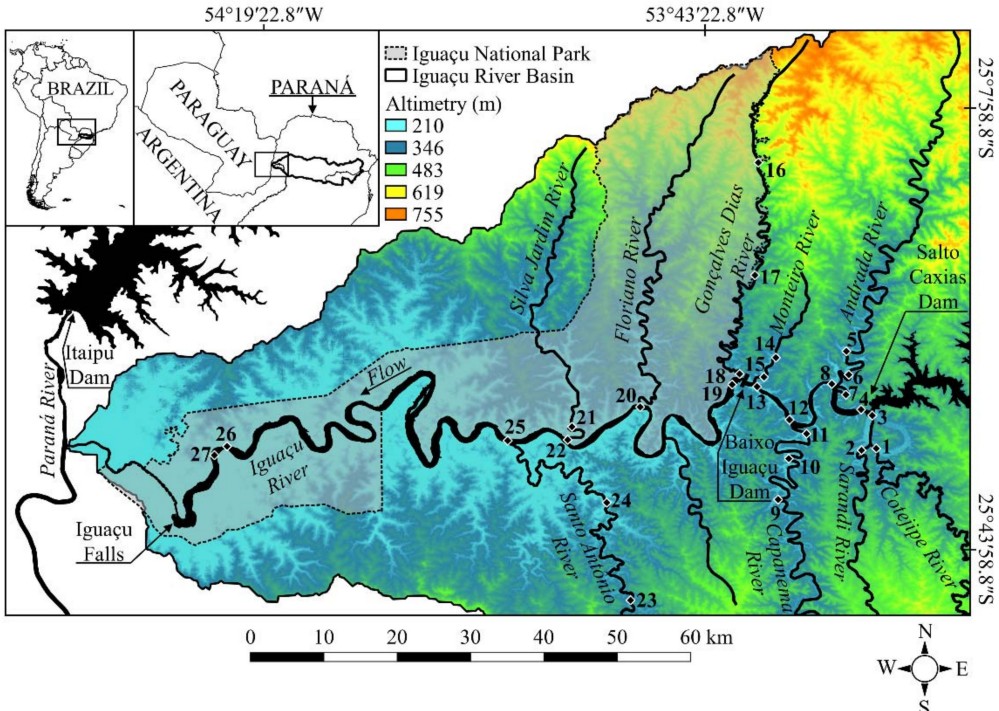

**Figure 1.** Sampling sites (1–27, diamonds outlined in white) located in the main channel of the Iguaçu River and main tributaries, from downstream of the Salto Caxias Dam to upstream of the Iguaçu Falls. Iguaçu National Park is highlighted in gray. QGIS software was used to elaborate the map.

## 2.2. Sampling and Data Analysis

2.2.1. Spatial Distribution and Abundance

Twenty-seven sampling sites were established, covering the main channel of the Iguaçu River and tributaries (Figure 1 and Table 1), to collect *S. melanodermatum* samples (Figure 2a). Samplings were undertaken monthly (70 samplings) from January 2010 to January 2011 (13 samplings, 25 locations) and from September 2012 to December 2016 (57 samplings, 27 locations). Some of these sampling sites were in the Iguaçu National Park Conservation Unit (16, 17, 18, 19, 20, 21, 22, 26, and 27) (Figure 2b and Table 1). Between sites 13 and 18, the new Baixo Iguaçu Dam was located. The sampling sites downstream of the Salto Caxias Hydroelectric Plant (1–15 and 22–24) exhibited anthropized areas (Figure 2c) due to livestock, agriculture, and degradation of riparian vegetation on both banks [43] (Figure 1 and Table 1). On the other hand, the sites that were within the boundary of the INP (16 to 20, 22, and 25) exhibited riparian vegetation that was well preserved in the right riverbank, while the left one impacted, except the sites 21, 26, and 27, which were fully within the INP with both margins being well preserved with dense forest (Table 1).

**Table 1.** Sampling sites' characteristics in the protected area of the Iguaçu National Park (INP) and outside the INP, namely, in the lower Iguaçu basin downstream of Salto Caxias Dam and upstream of Iguaçu Falls.

| Area | Environment | River | Site | Latitude | Longitude | Width of the River (m) |
|------|-------------|-------|------|----------|-----------|------------------------|
| Outside | Tributary | Cotejipe River | 1 | 25°35′17.04″ S | 053°29′56.58″ W | 25.4 |
| | | | 3 | 25°33′9.54″ S | 053°29′46.92″ W | 54.2 |
| Outside | Tributary | Sarandi River | 2 | 25°35′10.74″ S | 053°30′7.44″ W | 12.6 |
| Outside | Main channel | Iguaçu River | 4 | 25°32′30.18″ S | 053°30′37.98″ W | 373.2 |
| | | | 8 | 25°30′48.00″ S | 053°32′40.62″ W | 501.4 |
| | | | 12 | 25°31′2.28″ S | 053°32′34.44″ W | 586.9 |
| | | | 13 | 25°29′29.70″ S | 053°31′55.08″ W | 815.6 |
| Outside | Tributary | Andrada River | 7 | 25°27′36.18″ S | 053°31′51.69″ W | 120.4 |
| | | | 6 | 25°33′49.14″ S | 053°36′16.92″ W | 59.9 |
| | | | 5 | 25°34′16.26″ S | 053°35′52.68″ W | 35.6 |
| Outside | Tributary | Capanema River | 11 | 25°36′8.40″ S | 053°36′46.98″ W | 462.7 |
| | | | 10 | 25°39′54.84″ S | 053°37′15.66″ W | 82.7 |
| | | | 9 | 25°30′42.58″ S | 053°39′5.76″ W | 52.4 |
| Outside | Tributary | Monteiro River | 15 | 25°30′25.38″ S | 053°39′27.24″ W | 231.0 |
| | | | 14 | 25°28′12.96″ S | 053°37′39.00″ W | 9.2 |
| INP | Main channel | Iguaçu River | 19 | 25°29′57.54″ S | 053°40′53.52″ W | 750.7 |
| INP | Tributary | Gonçalves Dias River | 18 | 25°29′57.06″ S | 053°40′40.50″ W | 25.9 |
| | | | 17 | 25°21′48.12″ S | 053°39′18.00″ W | 63.5 |
| | | | 16 | 25°12′58.98″ S | 053°39′0.06″ W | 13.6 |
| INP | Tributary | Floriano River | 20 | 25°32′14.82″ S | 053°48′31.98″ W | 497.5 |
| INP | Tributary | Silva Jardim River | 22 | 25°34′51.24″ S | 053°54′43.68″ W | 26.6 |
| | | | 21 | 25°34′11.09″ S | 053°54′20.36″ W | 40.2 |
| Outside | Tributary | Santo Antônio River | 25 | 25°35′17.16″ S | 053°59′25.20″ W | 49.3 |
| | | | 24 | 25°40′25.80″ S | 053°51′15.90″ W | 37.0 |
| | | | 23 | 25°48′6.28″ S | 053°49′28.35″ W | 34.3 |
| INP | Main channel | Iguaçu River | 26 | 25°35′38.00″ S | 054°21′57.10″ W | 372.3 |
| | | | 27 | 25°37′13.20″ S | 054°23′29.20″ W | 932.6 |

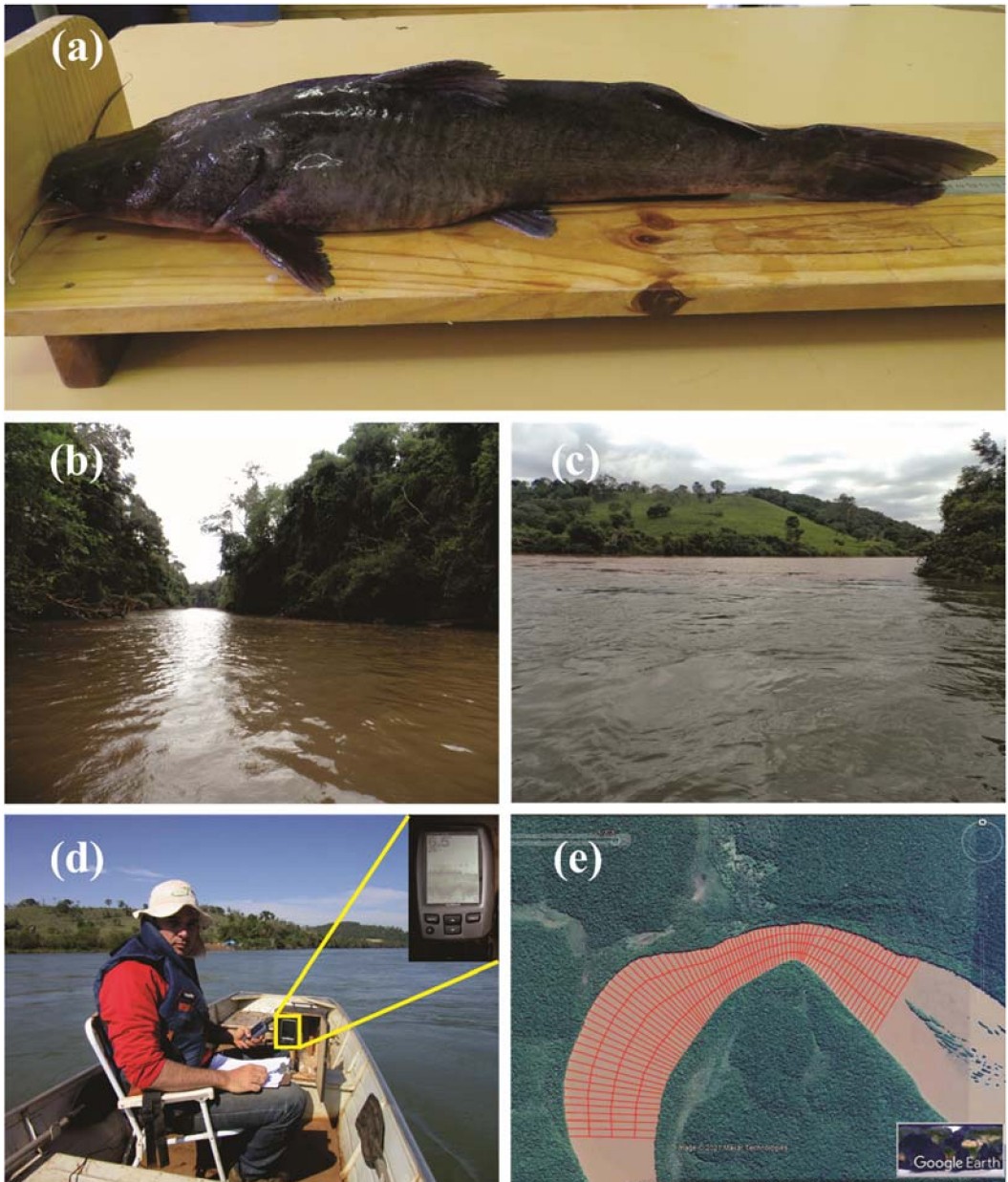

**Figure 2.** Specimen of *Steindachneridion melanodermatum* captured in the study area (**a**); sampling sites in the protected area at site 22 in Silva Jardim River (**b**) and outside of the protected area at site 8 in Iguaçu River (**c**); deep pools mapping (**d**,**e**).

The fishing gear used to catch the *S. melanodermatum* were gillnets (total length: 10 m and meshes: 2.5 to 18 cm between opposite nodes), trammel nets (total length: 10 m and meshes: 6, 7, and 8 cm between opposing nodes), and longlines (20 hooks of 9/0 and 20 hooks of 5/0 baited with pieces of fish and bovine heart). Nets and longlines were installed at 3 p.m. on the right bank of the river and inspected every six hours, that is, 9 p.m. that night and 3 a.m. and 9 a.m. of the next day, with a standardized fishing effort at all sampling sites.

The caught fish were measured (total and standard length) and weighed. Of the 180 specimens used in this study, 63 individuals were released and the others (117) were sacrificed for studies of their reproductive biology (already published [36]), diet, and genetic diversity (in preparation). Voucher specimens (eight specimens) were deposited at the Zoology Museum of the Universidade Estadual de Londrina (UEL) (MZUEL 15702, 15703, 15704, 17091, 17092, 17093, 17504, and 17531).

The fish were classified into juvenile and adult using the first maturation size of the species (juvenile: 21.0–43.9 cm) according to Assumpção et al. [36].

Fish abundance was expressed as catch per unit of effort (CPUE) for each piece of fishing equipment. The multigear mean standardization method (MGMS) [44] was used to standardize the efforts of the equipment.

Spatial abundance was assessed using non-parametric variance analysis (Kruskal–Wallis) to test for possible differences in the sampled sites (1 to 27), considering the areas (inside and outside of the Iguaçu National Park) and environments (main channel and tributaries). The CPUEs were transformed into a square root. The program STATISTICA v. 7.0 was used for this analysis.

### 2.2.2. Deep Pools Mapping

A hydroacoustic campaign was conducted in the study area to identify deep pools (notably, potential habitats of *S. melanodermatum*). Here we assumed that deep pools in all areas were those with depths lower than −5 m. Depths and geographical coordinates were obtained using sonar (Garmin Echo 100, Garmin International Inc., Olathe, KS, USA) and GPS (Garmin GPSMAP 60CX color map navigator, Garmin International Inc., Olathe, KS, USA), respectively (Figure 2d). In addition, the location of other deep pools in the Iguaçu River and tributaries was also provided by the Chico Mendes Institute for Biodiversity Conservation (ICMBio), Iguaçu National Park, regarding those of apprehension of predatory fishing of the species.

Bathymetric profiles were performed in deep pools (and around them) where the *S. melanodermatum* were caught. The transects were performed transversally to the main river channel (Figure 2e), with a spacing interval close to ten meters between the tracks [45]. To obtain the bathymetry data (river depths) and coordinates (longitudes and latitudes), we used a GPS (Garmin GPSMAP 60CX color map navigator) with sonar (Garmin Echo 100). The XYZ dataset that was collected in situ during the campaigns was processed using a geographic information system (GIS) to obtain the bathymetric digital terrain models (MDT); we implemented the triangular irregular networks (TIN) algorithm to represent the bathymetry surface. This interpolation approach is relevant for providing the spatial analysis of each deep pool in the study area.

The deep pools were classified according to their position in the river following Halls et al. [46]. Descriptions of the area and length of the pools, riparian vegetation, and habitat type (riffles, runs/rapids, pools) were also recorded. The area and length data were obtained based on Halls et al. [46]. The longitudinal extension of each pool along the thalweg was used to delineate the area, and the depth was digitized with 50 cm equidistance contours. A polygon was drawn manually for each pool in a geographic information system (GIS) by tracing the shallower depth contour crossed by the outlined pool line. Pool lengths, which were defined as the places where the riverbed profile crossed the zero-crossing line, were calculated as the distance along the thalweg between the start and end points.

The habitat type at each measurement point was subjectively assessed by the survey team [47]. The criteria for this assessment were: riffle—swiftly flowing with a high proportion of its water surface broken; pool—slow flowing with a smooth water surface; and rapids/runs—intermediate between pool and riffle with a wavy water surface.

## 3. Results

### 3.1. Spatial Distribution and Abundance

*Steindachneridion melanodermatum* were found in only 7 of the 27 sampling sites (4, 10, 11, 18, 25, 26, and 27), corresponding to 26% of the sampled sites. Considering those sites where the species occurred, three (43%) were located on the main channel of the Iguaçu River (sites 4, 26, and 27), and four (57%) were in tributaries (sites 10, 11, 18, and 25) (Figures 1 and 3).

A total of 180 specimens were caught with total lengths ranging from 21 to 102 cm (mean of 62.66 ± 15.36 cm SD) and body weight from 82.8 to 15,670.00 g (mean of 2905.93 ± 2411.25 g SD). Most fish sampled were adults—171 specimens, and 9 juveniles were captured. Of the total of adults, 98% (168 specimens) were sampled in the main channel (sites 4, 26, 27) and 2% (3 specimens) in the tributaries (sites 10, 11, 25). Regarding juveniles, 78% (seven specimens) occurred in the main channel (sites 26, 27) and 22% (two specimens) in the tributaries (sites 18, 25).

Fish were concentrated in the INP (1871.61 CPUE), highlighting the higher abundances in sites 26 and 27 (920.97 and 933.59 CPUE, respectively) (Figure 3), which contributed to the high CPUE recorded in the main channel of the Iguaçu River (1925.32 CPUE). On the other hand, low abundance values were recorded outside the INP (115.38 CPUE) and in tributaries (61.67 CPUE). The species were found downstream of Salto Caxias Dam in the main channel at site 4 (70.76 CPUE), and three tributaries: Capanema, outside of the INP, site 10 and 11 (18.37 CPUE); Gonçalves Dias River, the border with INP, site 18 (17.05 CPUE); and Santo Antônio River, outside of the INP, site 25 (26.24 CPUE) (Figure 3). Significant differences were observed between the CPUE values in the areas (inside and outside of the INP) (H (1, N = 324) = 31.69; $p < 0.01$) and the environments (main channel and tributaries) (H (1, N = 324) = 61.73719; $p < 0.01$) sampled.

The greatest abundance of *S. melanodermatum* was registered at site 26, a place with a high depth (22 m) (Figure 3). A high abundance also occurred at site 27, possibly because of its proximity to deep sites. The capture of the species was also observed downstream of Salto Caxias Dam at site 4 with a 25 m depth, and in the Capanema River, a tributary with a maximum depth of 14 m (Figure 3).

### 3.2. Deep Pools Characterization

Twenty-four high-depth sites were identified in the study area, with maximum values ranging from 5 to 25 m (Figure 4). These sites were called deep pools, especially P1 and P23 with the greatest depths (25 and 22 m, respectively). Most of the deep pools (21) were along the main channel of the Iguaçu River and 3 were in tributaries: Capanema (1), Floriano (1), and Santo Antônio (1) Rivers (Figure 4); 63% of the deep pools were in the protected area and 37% were outside of the protected area. From all the deep pools mapped in the main channel of the Iguaçu River, it was possible to determine the depths in 16 sites, with maximum values ranging from 13 to 25 m (P1, P3–P5, P7–P13, P15–P17, P23), and 1 in the Capanema river with a depth of 14 m (P6) (Figure 4b). Considering the deep pools indicated by ICMBio regarding predatory fishing data, five were in the main channel of the Iguaçu River at INP (P19–P22) and one was outside the INP (P18), in the Santo Antônio River (border Brazil and Argentina) (Figure 4a—sites indicated using white squares). However, depth measurements were not performed in these sites, mainly due to the low river level and intense rapids, making navigation impossible. According to ICMBio (and Lucileine Assumpção's personal observations), these deep pools exhibited depths greater than 5 m.

The deep pools identified in the main channel of the Iguaçu River and tributaries were classified into three main types in terms of their positions (Figure 5). Of the 24 deep pools, 58% (15 pools—P1, P4, P5, P10–P13, P15, P17–P19, P21–P24) were in the meanders of the Iguaçu River (Figure 5a), 34% (9 pools—P2, P3, P6–P9, P14, P16, P20) in the middle of the main channel (Figure 5b), and 8% (2 pools—P19, P23) downstream of mid-channel island (Figure 5c). The deep pool P23, known as Poço Preto, was in the meander of the Iguaçu River and downstream of the mid-channel island, namely, Taquaras' Island. Pools P12 and P13 in the Iguaçu River main channel were located close to the confluence with the Floriano River: P12 was located approximately 1252 m upstream of the confluence, and P13 was 533 m downstream of the confluence.

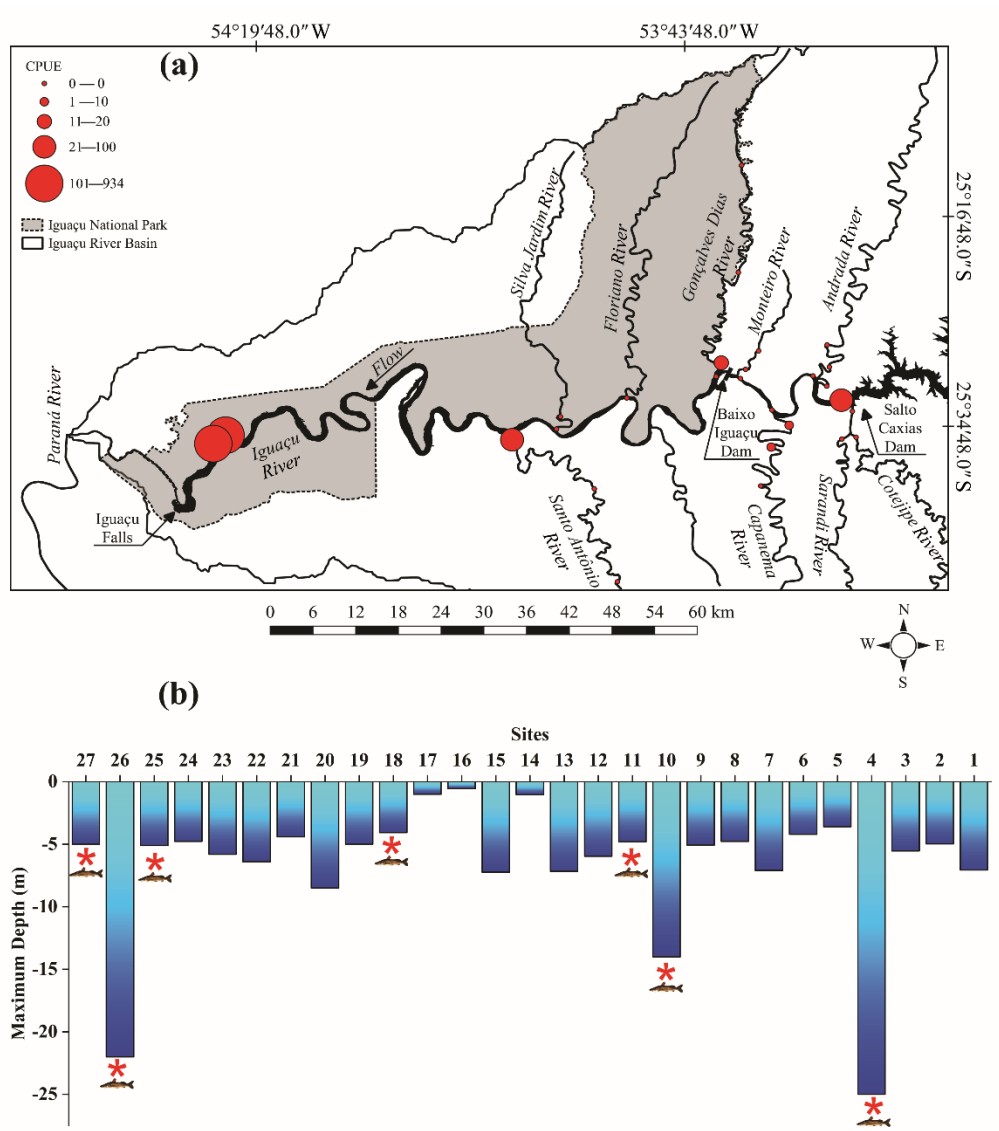

**Figure 3.** Spatial distribution of *Steindachneridion melanodermatum* with respective abundances (CPUE in the red bubble) (**a**) and maximum depths (**b**) of sampling sites that were examined in Lower Iguaçu, from downstream of the Salto Caxias Dam to upstream of Iguaçu Falls. Asterisks (*) indicate sites where the species were found.

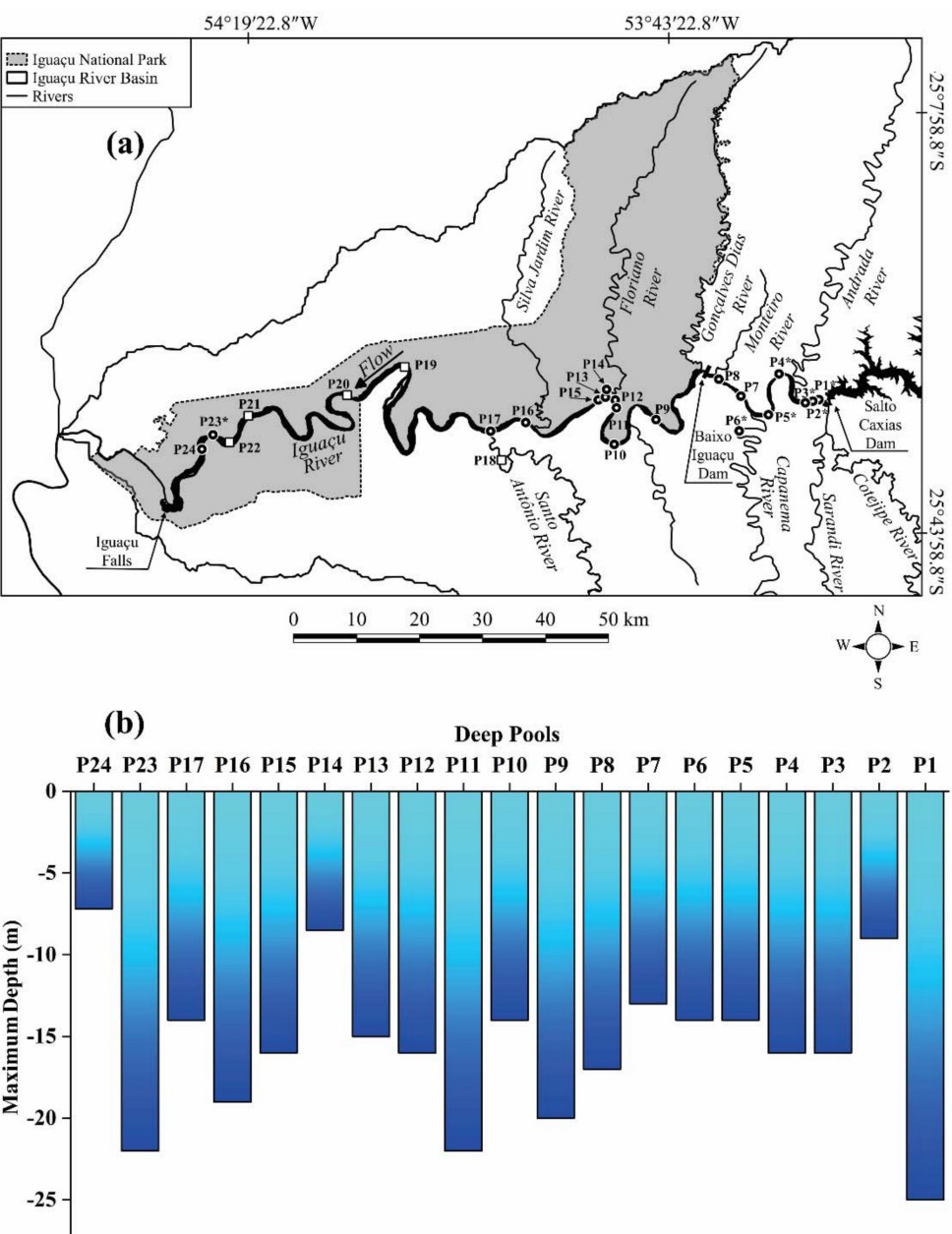

**Figure 4.** Locations of deep pools (**a**) and their maximum depths (**b**) in the Lower Iguaçu River, downstream of the Salto Caxias Dam, upstream of the Iguaçu Falls. Iguaçu National Park (INP) is highlighted in gray. □: deep pools indicated by ICMBio. Asterisks (*) indicate where the bathymetry was performed to characterize the deep sites.

In addition to variations in depths of the deep pools P1 to P24, data on the area, length, riparian forest, and habitat type are shown in Table 2. The width of the transverse transect of the bathymetric profile is exhibited for deep pools and the surrounding area where the species were found (P1, P2, P3, P4, P5, P6, and P23) (Figure 6 and Table 2).

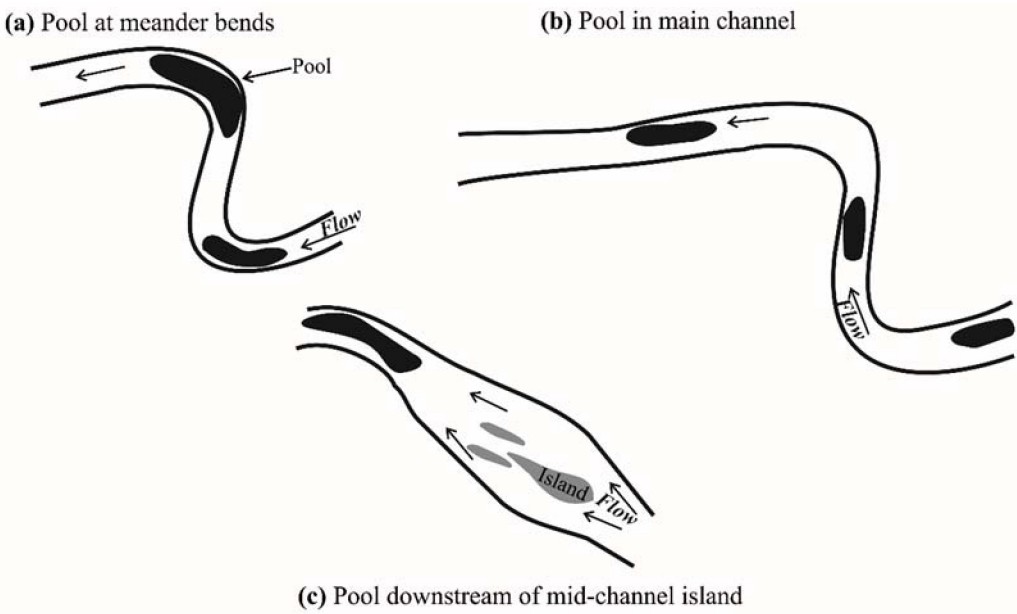

**Figure 5.** Main deep pool types in the Iguaçu River and tributaries downstream of the Salto Caxias Dam, upstream of the Iguaçu Falls in Iguaçu National Park. Adapted from Halls et al. [46].

The riparian zone in the main channel of Iguaçu River, pools P1 to P5 and P7 to P8, and Capanema River, P6, which were all located outside of the protected area (INP), exhibited riparian forest that had been reduced to tiny fragments on one or both banks occupied by agricultural activities and pastures (Figure 7 and Table 2). In the protected area, the riparian zone exhibited dense preserved native forest on both riverbanks, as in pools P23 (Figure 7), P20 to P22, and P24 (Table 2), or just on one of the riverbanks (P9 to P17 and P19) (Table 2).

The total area of the sampling transects for mapping the pools ranged from 1.253 to 258.39 ha (Table 2). Deep pools had a low initial depth, varying from 0.40 to 2.0 m, then increasing abruptly, with maximum depths of 5.0 to 25 m (Figures 6 and 7 and Table 2). The bathymetric transects conducted (P1, P2, P3, P4, P5, P6, and P23) had areas ranging from 2.06 (P6) to 258.39 (P23) ha that housed deep pools with areas ranging from 0.451 to 47.255 ha, with an average depth of 6.98 m, a maximum depth of 25.0 m, and different bathymetric profiles (Figures 6 and 7 and Table 2).

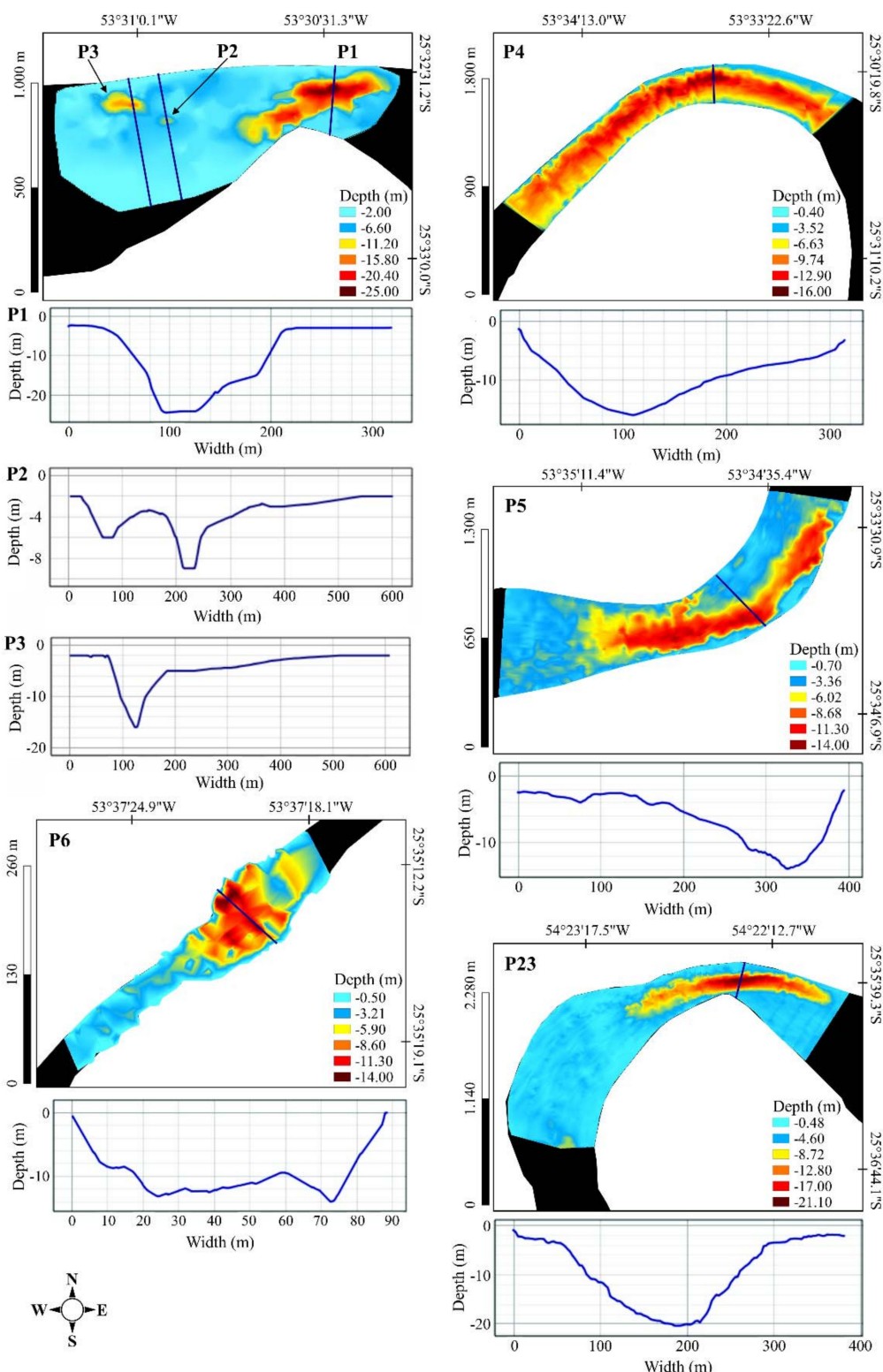

**Figure 6.** Map and bathymetric profile of the margin-to-margin deep pools' transverse transects in the main channel of the Iguaçu River outside of the INP (P1, P2, P3, P4, P5) and inside of the INP (P23), and a tributary, namely, Capanema River (P6), outside of the INP. The bathymetric profile of the cross-section (blue line) corresponds to the portion with the greatest depth.

**Table 2.** Data of depth variations (min: minimum; max: maximum), location of deep pools (MC: main channel; TR: tributary), deep pool positions (a: pool at meander; b: pool at main channel; c: pool downstream of mid-channel island), sampling area, transect width, and main characteristics of the deep pools from downstream of the Salto Caxias Dam to upstream of the Iguaçu Falls. The general sampling areas of pools P1, P2, and P3 were the same, as these pools were very close. Asterisks (*) indicate where the bathymetry was performed to characterize the deep pools.

| Deep Pool | Depth (m) | | Location | Position | Area (ha) | Deep Pool Area (ha) | Deep Pool Length (m) | Transect Width (m) | Characteristics |
|---|---|---|---|---|---|---|---|---|---|
| | Min | Max | | | | | | | |
| P1 * | 2.4 | 25.0 | MC | a | 67.06 | 10.186 | 766.511 | 647.03 | Rocky bed, large rocks, and rapids. The river's right and left banks had a greater concentration of pastures and few forest fragments (Figure 7). |
| P2 * | 0.4 | 16.0 | MC | b | 67.06 | 0.451 | 111.542 | 605.456 | Rapids with a riparian zone that was poorly preserved with forest fragments and a predominance of pasture on both banks (Figure 7). |
| P3 * | 0.9 | 14.0 | MC | b | 67.06 | 1.847 | 277.006 | 615.277 | Rapids with a riparian zone that was poorly preserved with forest fragments and predominance of pasture on both banks (Figure 7). |
| P4 * | 0.4 | 16.0 | MC | a | 93.33 | 47.255 | 2994.951 | 387.32 | Rapids with the left bank impacted by agricultural activities and almost nonexistent riparian forest on both banks (Figure 7). |
| P5 * | 0.9 | 14.0 | MC | a | 87.91 | 31.559 | 1781.068 | 474.60 | Some stretches with low depths, riffles, where the navigation was impossible due to turbulence, other rapids stretches, rocky outcrops, and the presence of a small island in the middle of the river. Human activities, such as agricultural areas and pastures, on both banks (Figure 7). |
| P6 * | 0.5 | 14.0 | TR | b | 2.06 | 1.046 | 244.265 | 60.66 | Riffles upstream of the pool. Human activities, such as agricultural areas, on the right bank, with backwaters in the pool. |
| P7 | 1.0 | 13.0 | MC | b | 85.281 | - | - | - | Rapids stretches near riffles, with an approximate width of 416 m. Both banks with a predominance of pastures and agricultural areas, and a riparian forest that was barely preserved. |

**Table 2.** *Cont.*

| Deep Pool | Depth (m) | | Location | Position | Area (ha) | Deep Pool Area (ha) | Deep Pool Length (m) | Transect Width (m) | Characteristics |
|---|---|---|---|---|---|---|---|---|---|
| | Min | Max | | | | | | | |
| P8 | 1.0 | 17.0 | MC | b | 23.559 | - | - | - | Rapids stretches located close to shallow water areas, namely, rifles, with exposed rocks before and after the stretch, with an approximate width of 290 m. Right bank had a small section of riparian vegetation that was barely preserved. On the left bank, there were pasture and agricultural areas. |
| P9 | 2.0 | 20.0 | MC | b | 108.833 | - | - | - | Rapids stretch located upstream of an island, with strong running areas and an approximate width of 280 m. Right bank had a predominance of preserved riparian forest. On the left bank, there were pasture, agricultural areas, and small forest fragments that were barely preserved. |
| P10 | 1.5 | 14.0 | MC | a | 1.253 | - | - | - | Rapids stretches with riffles, shallow water, and exposed rocks, as well as a pool that was located in a steep meander, upstream to an island in the middle of the river, with an approximate width of 78 m. Right bank had a predominance of preserved riparian vegetation. On the left bank, there were pasture, agricultural areas, and small forest fragments that were barely preserved. |
| P11 | 1.5 | 22.0 | MC | a | 27.010 | - | - | - | Rapids stretch upstream of an accentuated meander, with an approximate width of 281 m. The right bank had a predominance of preserved riparian forest. On the left bank, there were pasture, agricultural areas, and small forest fragments that were barely preserved. |
| P12 | 1.5 | 16.0 | MC | a | 46.628 | - | - | - | Rapids stretch in a meander with an approximate width of 320 m. The right bank exhibited dense riparian forest. The left bank had a predominance of agricultural area and small forest fragments that were barely preserved. |

**Table 2.** *Cont.*

| Deep Pool | Depth (m) | | Location | Position | Area (ha) | Deep Pool Area (ha) | Deep Pool Length (m) | Transect Width (m) | Characteristics |
|---|---|---|---|---|---|---|---|---|---|
| | Min | Max | | | | | | | |
| P13 | 1.0 | 15.0 | MC | a | 41.399 | - | - | - | Rapids stretch in a meander with an approximate width of 409 m. The right bank exhibited dense riparian forest. The left bank had a predominance of agricultural area and small forest fragments that were barely preserved. |
| P14 | 1.0 | 8.5 | TR | b | - | - | - | - | Rapids stretch with a dense riparian forest that was preserved on both banks. |
| P15 | 1.5 | 16.0 | MC | a | 97.655 | - | - | - | Rapids stretch upstream of a meander, with an approximate width of 319 m. The right bank had a predominantly riparian forest that was preserved. The left bank presented an agricultural area and small forest fragments that were barely preserved. |
| P16 | 1.0 | 19.0 | MC | b | 176.230 | - | - | - | Rapids stretch that was approximately 268 m in width and located in a meander. The right bank had preserved riparian vegetation. The left bank presented a predominantly agricultural area with poorly preserved forest fragments. |
| P17 | 1.0 | 14.0 | MC | a | 46.976 | - | - | - | Rapids stretch in a meander with an approximate width of 335 m. The right bank had preserved riparian vegetation. The left bank presented a higher concentration of pastures with shrubby vegetation, an agricultural area, and small forest fragments that were barely preserved. |
| P23 * | 0.8 | 22.0 | MC | a, c | 258.39 | 43.693 | 2303.019 | 153.27 | Riffle stretches in some shallow areas, bedrock substrate forming extensive areas with large rocks, backwater areas that included foams on the water surface, and an island upstream of the pool (Taquaras' Island). The deep pool was in a meander. Both banks had dense riparian forests (Figure 7). |
| P24 | 4.07 | 7.2 | MC | a | - | - | - | - | Intense rapids stretch with an approximate width of 927 m. Stretches of flooded areas on the left bank and preserved dense riparian forest on both banks. |

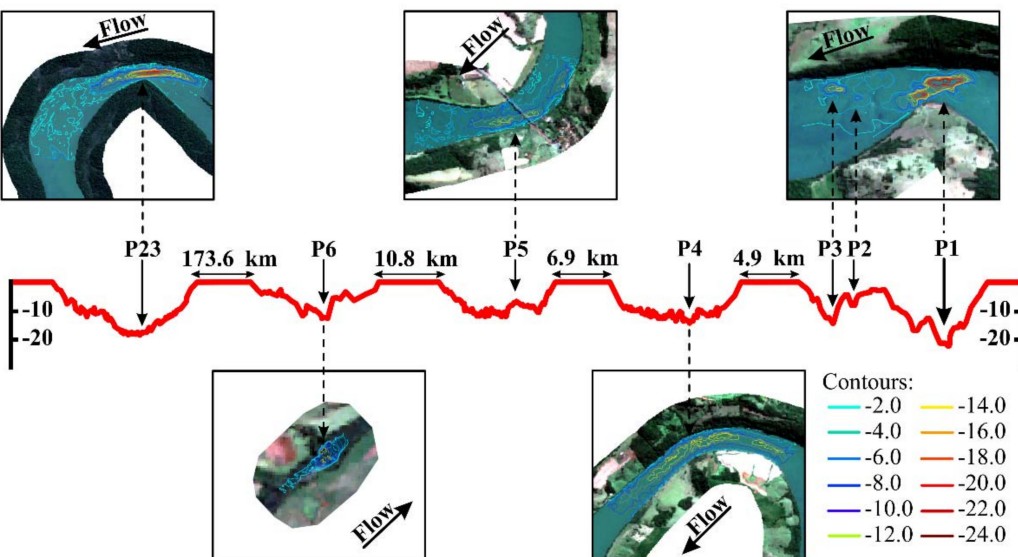

**Figure 7.** View of the riverbanks and general longitudinal profile of the deep pools located outside the protected area (INP) (P1, P2, P3, P4, and P5 in the main channel of the Iguaçu River, and P6 in the tributary, namely, the Capanema River) and in the protected area (INP) (P23). Horizontal arrows indicate the distances between the pools. Continuous vertical arrows indicate the places of greater depths. Dashed vertical arrows indicate the location of the pool. The red line indicates the section of depths.

## 4. Discussion

Protected areas are one of the main tools used for the biological conservation of aquatic habitats and species worldwide [48–51], positively affecting fish productivity in freshwater environments [48]. The higher abundance of *S. melanodermatum* in the protected area was different from the unprotected area downstream of the Salto Caxias Dam to the upper limit of the Iguaçu National Park. Habitat changes and degradations caused by the hydroelectric plant (downstream of Salto Caxias Dam), high anthropic pressure (agriculture and pasture), and illegal and predatory fishing [9,22,36] along the main channel and tributaries may have contributed to the low abundance in this area. Studies carried out on Lake Tanganyika in Tanzania to assess and compare differences in fishing resources within protected areas and outside (unprotected areas) have also shown that species abundance and diversity were more significant in the protected area [52]. The abundance and diversity of vulnerable and endangered species were greater in the protected areas than the unprotected ones [9,53], showing the benefit of the conservation of the large catfish that are endangered in the Lower Iguaçu River basin.

Our study revealed that *S. melanodermatum* inhabited deep pools (high-depth sites) in the Lower Iguaçu River and tributaries. The deep pool called Poço Preto (P23) and its surroundings in the Iguaçu National Park can be considered a sanctuary for the species given the highest abundances. Thus, preserving this area is critical for the conservation of this large catfish that is threatened by human actions. Identifying sites that are essential to the conservation of freshwater species, together with information about critical species, improves the representation of freshwater biodiversity in protected areas [28,54].

Deep pools are of high ecological importance for conserving tropical fish species [55]. These locations serve as refuge habitats in the dry season for sedentary and migratory species in the Mekong River [56–61], as well as permanent habitats for other species, such as non-native species [61], and as spawning and feeding sites [38,56]. Deep pools are fundamental for sustaining the Lower Mekong River basin fisheries, providing critical spawning and refuge habitats for nearly 200 species of fish, including the Mekong giant catfish *Pangasianodon gigas* and other critically endangered fish species [46]. In addition,

deep pools are an essential link between habitats for migratory fish species migrating over long distances, which use the pools for rest [56,62]. In dry seasons, pools and rapids can become an obstacle to fish trying to swim through them [63].

Considering that habitats with deeper areas attract larger fish [64], *S. melanodermatum*, the unique large species in the Iguaçu River basin, may depend on deep pools as a refuge habitat during periods when the river level is low during dry periods. Congenerous species also use deep sites, such as *Steindachneridion parahybae* from Paraíba River [65] and *S. scriptum* from Uruguai River [66–69]. In the Paraguay River, the existing pools are used as a refuge by several fish species, such as large pimelodids, especially the jaú (*Zungaro zungaro*, cited as *Paulicea luetkeni*) when the river level goes down [70]. During the water level elevation, the free movement of fish is possible, allowing them to select habitats, while at low levels, there is a limitation of access to pools or the main channel in large rivers [71]. On the Iguaçu River, drastic water level fluctuations occur downstream of the Salto Caxias Dam, which are caused by the hydroelectric operations during the week in the early hours of the morning and on the weekends throughout the day. Therefore, the identification of pools by fishers makes it easier to capture the *S. melanodermatum*, as in P1, which was a deep pool located close to the Salto Caxias Dam. In this sense, some deep pools in the Iguaçu River that are used as refuge sites become traps that are the targets of illegal fishing of *S. melanodermatum*.

This study is the first to describe deep pools in the Lower Iguaçu River basin, downstream of the Salto Caxias Dam to Iguaçu Falls. The Uruguay River also has stretches with waterfalls, followed by deep pools with depths from 20 to 30 m and bedrock with depths of 1 to 2 m [63]. In addition to their depth, deep pools differ in environmental characteristics, such as water velocity and type of substrate [72], and are classified according to their position on the riverbed, presence of islands, and substrate [46]. In our study, most deep pools were positioned in the meander and the middle of the main channel, with bedrock dominating and occupying small to large areas. Halls et al. [46] highlighted that pool length, area, and volume tend to increase with distance downstream, suggesting that discharge (which increases with growing catchment area) plays a key role in determining overall pool size. In large rivers worldwide, hydroelectric dams have altered the natural flow regime [73–75] and impacted sediment transport [73,74]. Reservoirs in the Upper Volga River have decreasing bottom irregularity and the resulting loss of efficiency of its use by fish for feeding and protection against predators, leading to the decline of rheophilic fish species [76]. In addition, the Volga cascade of dams caused the disappearance of iconic species, such as the Caspian lamprey (*Caspiomyzon wagneri*), Caspian inconnu (*Stenodus leucichthys*), Russian sturgeon (*Acipenser gueldenstaedtii*), and the Beluga (*Huso huso*) in the Upper Volga [75]. Hydroelectric dams change the water flow and transport sediments, threatening the deep pools [46]. The upcoming loss of some deep pools downstream from the Salto Caxias Dam reveals the severe impact that the new dam may have on the preferential habitat of *S. melanodermatum*. These pools could become silted up through high sedimentation rates over the years, which seriously harms many fish, particularly species that are dependent on deep-water areas for their survival [57,77]. Sedimentation promotes the fragmentation of aquatic habitats, which affects the health and production of fish and reduces primary production [18]. Given this context, we emphasize that *S. melanodermatum* and other fish species inhabiting deep pools in the main channel of the Iguaçu River could be imperiled.

The Mekong River and its tributaries are considered the last refuges for the megafauna of large fish at high risk of extirpation because of anthropogenic pressures, requiring political actions that include inspection for the conservation of these species in their natural habitat [26]. The occurrence of the *S. melanodermatum* in the main channel and tributaries of the Iguaçu River reveals their importance in the conservation of species that use deep pools as preferential habitats. Some studies report that tributaries are critical for spawning and recruiting many fish species, especially those that migrate [78–80]. The tributaries hold endangered and threatened species and supply various environmental conditions,

including access to spawning habitats that provide refugia for early life stages [80]. The protection area needs to enable young individuals to reach sexual maturity within that area to efficiently sustain a fish population while maintaining the diversity of fish species regardless of fishing pressure [81]. The prevalence of adults of *S. melanodermatum* in the deep pools located in the main channel may indicate that tributaries are an environment for development and growth, and spawning areas of this species still need to be elucidated by identifying the presence of eggs and early larvae [36]. As there is no knowledge regarding movement behavior, spawning sites, and initial development areas of *S. melanodermatum*, it is essential to protect and preserve not only the Iguaçu River, but also its tributaries, such as the Capanema, Floriano, Gonçalves Dias, and Santo Antônio Rivers, where the species was recorded.

Deep pools were extensively studied and mapped to protect and conserve the fish species of the Mekong River, with the establishment of Fish Conservation Zones (Fisheries Conservation Zones (FCZs)) or sanctuaries, where fishing is not allowed [38,72], with direct effects on the increase in the abundance of several species, both sedentary and migratory [56]. In this way, protected areas benefit the conservation of freshwater fish [48,53,82]. The Poço Preto (P23) deep pool and its surroundings, located in the protected area of Iguaçu National Park, were found to be fundamental for the maintenance of *S. melanodermatum* since downstream of the Salto Caxias Dam, several anthropic actions further expose the species to threats. This survey and delimitation of priority areas for the spatial distribution of the endemic species highlight the need to create new conservation units in areas of their occurrence to minimize population losses and ensure the protection of aquatic biodiversity. Based on this information, our results indicate where *S. melanodermatum* are present, particularly showing which are the locations with the highest abundances, and it is also essential to consider that the low abundance of *S. melanodermatum* in the downstream stretch of the Salto Caxias reservoir are the results of the high local anthropic action. Thus, the definition and creation of environmental protection areas become essential for maintaining and conserving this species.

Establishing areas for habitat protection, such as sanctuaries, protected areas for fish, or prohibiting fishing in certain regions, are measures that have been used recently for habitat conservation for some endemic fish species [26,83]. Therefore, we recommend that Poço Preto (deep pool P23, location 26) and the deep pools located downstream of the Salto Caxias Dam and in the Capanema River, a tributary of the new reservoir and a target of new hydroelectric projects, are established as conservation areas and areas where fishing is prohibited (with intense inspection). The studies carried out by Assumpção et al. [9,36] emphasized the importance of the protected area, namely, the Iguaçu National Park, as well as Poço Preto for the maintenance and conservation of *S. melanodermatum* and other endemic species in the Iguaçu River basin, highlighting this region as a sanctuary for the Iguaçu River's ichthyofauna. Mitigation and environmental compensation programs should not only restrict lentic sites but the management carried out should address fishery production and aim to conserve biological diversity [8], especially endemic and endangered species.

The greatest abundance of the *S. melanodermatum* occurred in sites that bordered Argentina in the Iguaçu National Park, which is also a protected area. Studies carried out in the INP in the Argentina border also registered the species in Porto Três Marias, in the Garganta do Diabo walkway, and in the mouth of the Ñandú River, which are all locations near the Iguaçu Falls [84]. Considering that our study area borders Argentina, it is of extreme importance for the environmental agencies of both countries to elaborate specific strategies for the conservation of *S. melanodermatum* and come to a joint agreement for the inspection of the illegal fishing that occurs in the pools in the INP. The management of transboundary waters is necessary, and the fisheries face challenges that need to be managed [25].

Humans have altered natural river connectivity in multiple ways by placing structures in the longitudinal or lateral flow paths, such as dams and levees [85]. Dams and reservoirs, along with their upstream and downstream propagation of fragmentation and flow regula-

tion, are the leading contributors to the loss of river connectivity [85,86]. The construction of new hydroelectric power plants, such as the Baixo Iguaçu HPP hydroelectric power plant that is in operation, will result in the loss of habitats and restrict the area of occurrence of the *S. melanodermatum*, which may cause a decrease in genetic diversity via fragmentation. Endemic species are particularly vulnerable to environmental changes, increasing the risk of extinction [87]. *Steindachneridion melanodermatum* is becoming rare and difficult to capture, especially in non-protected areas downstream of the Salto Caxias Dam, where our captures were low. Although it is distributed in these deep pools, unfortunately, it was impossible to make correlations due to the scarcity of fish sampled. Thus, the new Baixo Iguaçu HPP caused the reduction of the dam-free stretch in the Iguaçu River and prevented *S. melanodermatum* from accessing the deep pools upstream (8) and downstream (16) of the new dam in the main channel and tributaries. This species is seriously threatened with a risk of disappearing if conservation measures are not established [36] in its last refuge. Because of developing new monitoring technologies and the current push for more robust national and international mechanisms for biodiversity management, threatened fish should no longer be neglected in conservation and sustainability commitments.

## 5. Conclusions

Given watersheds' current status and prospects, actions are needed to protect these threatened river systems, which provide habitats for endemic, rare, and endangered species, such as *S. melanodermatum*. In the Lower Iguaçu River, special attention should be given to the main channel and tributaries downstream of Salto Caxias Dam, upstream of Iguaçu Falls, and in protected areas of the last free-flowing river stretch. *Steindachneridion melanodermatum* prefers deep pools, but its greater abundance occurred in entirely preserved INP regions, evidencing the importance of these sites in conserving the species. Therefore, the leading conservation strategies that are required to safeguard *S. melanodermatum* are: (i) establishing deep pools as ecological sanctuaries, (ii) the intensification of illegal fishing inspection by environmental agencies, and (iii) the maintenance of dam-free tributaries. In addition, evaluating and monitoring the transport of sediments, silting, and the *S. melanodermatum* population in the deep pools using hydroacoustic cameras are mandatory to support the management and conservation of this species.

**Author Contributions:** Conceptualization, L.d.A., M.C.M., E.A.L.K., E.G., L.S.-S., O.A.S., and S.M.; methodology, L.d.A., M.C.M., J.F.M.d.S., K.A.S.d.M., S.F.R.P., P.S.d.S., E.G., and S.M.; formal analysis, L.d.A., J.F.M.d.S., K.A.S.d.M., S.F.R.P., P.S.d.S., and E.A.L.K.; data curation, L.d.A., M.C.M., and S.M.; writing—original draft preparation, L.d.A., M.C.M., J.F.M.d.S., E.G., and S.M.; writing—review and editing, L.d.A., M.C.M., E.A.L.K., E.G., L.S.-S., O.A.S., and S.M.; supervision, L.d.A., M.C.M., and S.M.; project administration, L.d.A., M.C.M., and S.M.; funding acquisition, M.C.M. and S.M. All authors have read and agreed to the published version of the manuscript.

**Funding:** The Macuco Safari and Consórcio Empreendedor Baixo Iguaçu (CEBI) funded this research.

**Institutional Review Board Statement:** All fieldwork for fish sampling complied with the legal regulations of Brazil. The collection licenses were granted through the Authorizations of the Environmental Institute of Paraná-IAP (License n° 37788 and n° 43394), by the Chico Mendes Institute for Biodiversity Conservation—ICMBio (n° 003/2014 and Official SEI n° 63/2016-DIBIO/ICMBio), and by the Biodiversity Authorization and Information System (SISBIO) (n° 25648-3 and 25648-4). The procedures used were approved by the Ethics Committee on the Use of Animals—CEUA of the Universidade Estadual do Oeste do Paraná (Protocol code 62/09; November 01, 2009).

**Informed Consent Statement:** Not applicable.

**Data Availability Statement:** The data that support the findings of this study are available from the corresponding author upon reasonable request.

**Acknowledgments:** We thank ICMBio/Iguaçu National Park for their logistical support and the opportunity to develop this study. Additionally, we thank Instituto Água Viva for help with the logistics and the technical team of the Grupo de Pesquisa em Tecnologia em Ecohidráulica e Conservação de Recursos Pesqueiros e Hídricos (GETECH), for assisting in the field samplings: Pércimo

Noronha Chiaretto, Dhonatan Oliveira dos Santos, and Fabio Luiz Paetzhodt. Coordination for the Improvement of Higher Education Personnel (CAPES) provided a doctoral scholarship to Lucileine de Assumpção, and the National Council for Scientific and Technological Development (CNPq) provided a Productivity Grant in Technological Development and Innovative Extension (DT) to Sergio Makrakis.

**Conflicts of Interest:** The authors declare no conflict of interest.

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
