# Peer review of "Deep Pools: Ecological Sanctuaries for Steindachneridion melanodermatum, a Large Endemic and Endangered Pimelodid of the Iguaçu River"

_water, doi:10.3390/w13121700_

Round 1

Reviewer 1 Report

  1. Abstract. It is necessary to supplement the abstract with a few more specific conclusions.
  2. introduction. Line 51-55. Specify other references on this topic (for example, http://dx.doi.org/10.24189/ncr.2018.048; https://doi.org/10.15560/13.4.185; https://dx.doi.org/10.24189/ncr.2019.036).
  3. Materials and Methods. You must specify what was done with the material. The species is rare and its catch with subsequent killing is unacceptable. How many fish were handed over to the museum and how many were released?
  4. Results. Acceptable.
  5. Discussion. In the discussion, it is desirable to add additional references on rare fish species, on studies of rare species in protected areas. Here are examples of such references: http://dx.doi.org/10.24189/ncr.2019.018; https://doi.org/10.1186/s12898-019-0258-4; http://dx.doi.org/10.24189/ncr.2018.043; https://doi.org/10.1111/cobi.13466; http://dx.doi.org/10.24189/ncr.2018.041; DOI: 10.1134/S1067413609030084. It is necessary to give examples of publications that describe how power plant dams contribute to reducing river flow, silting up the bottom and reducing the number of bottom fish species. Such examples are well known in the Volga River in Russia. Also explain in what places this species breeds? In what places of the river are the spawning areas of S. melanodermatum? How are juveniles and adults distributed in the river?
  6. Conclusions. Acceptable.

Author Response

Dear Reviewer,

We agree with the comments and suggestions, and we did a review of the manuscript (Water – 1253080) according to requests. The corrections and adjustments are marked up using the "Track Changes" function. The comments and our responses to them are below.

We thank the comments and suggestions to the manuscript, which contributed to its improvement.

Yours sincerely,

Lucileine de Assumpção
Universidade Estadual do Oeste do Paraná

---------------------------------------------------------------------------------------

Responses to reviewer

Reviewer #1 - Comments and Suggestions for Authors

 1. Abstract. It is necessary to supplement the abstract with a few more specific conclusions.

RESPONSE: We improved the conclusions in the abstract.

2. Introduction.

Line 51-55. Specify other references on this topic (for example, http://dx.doi.org/10.24189/ncr.2018.048; https://doi.org/10.15560/13.4.185; https://dx.doi.org/10.24189/ncr.2019.036).

RESPONSE: We inserted some references.

3. Materials and Methods.

You must specify what was done with the material. The species is rare and its catch with subsequent killing is unacceptable. How many fish were handed over to the museum and how many were released?

RESPONSE: Most of the 180 specimens of S. melanodermatum used in our study are the same ones used in the study of the reproductive biology of the species published in Assumpção et al. (2021), entitled "Population structure and reproduction of Steindachneridion melanodermatum (Siluriformes: Pimelodidae), a large endemic catfish to Neotropical ecoregion", Mar. Freshw. Res., 2021, https://doi.org/10.1071/MF19373. In addition, two more studies (diet and genetics) are being developed with these same individuals. Therefore, 180 specimens were sampled in the study, but 63 were released, and 117 killed for laboratory analysis (08 deposited in the museum). The specimens were captured and killed under the permission of the Federal Agency of Nature Conservancy (ICMBio) and the Animal Experimentation Ethics Commission. We have added this information in the Material and Methods section of the manuscript for further clarification.

4. Results. Acceptable.

5. Discussion.

In the discussion, it is desirable to add additional references on rare fish species, on studies of rare species in protected areas. Here are examples of such references: http://dx.doi.org/10.24189/ncr.2019.018; https://doi.org/10.1186/s12898-019-0258-4; http://dx.doi.org/10.24189/ncr.2018.043; https://doi.org/10.1111/cobi.13466; http://dx.doi.org/10.24189/ncr.2018.041; DOI: 10.1134/S1067413609030084.

RESPONSE: We added references about these issues in the discussion.

 -It is necessary to give examples of publications that describe how power plant dams contribute to reducing river flow, silting up the bottom and reducing the number of bottom fish species. Such examples are well known in the Volga River in Russia.

RESPONSE: We included them.

-Also explain in what places this species breeds? In what places of the river are the spawning areas of S. melanodermatum?

RESPONSE: The study on the reproductive biology of the sampled specimens, conducted by Assumpção et al. (2021), showed specimens capable of spawning and post-spawning, especially in the main channel of the Iguaçu River, P23 and surroundings, in the protected area near Iguaçu Falls. Additionally, juveniles also occurred at these sites and in the tributaries. Tributaries may play an important role in the development and growth of S. melanodermatum. However, this species' spawning areas still need to be investigated through the occurrence of eggs and early larvae. We inserted this information into the discussion.

-How are juveniles and adults distributed in the river?

RESPONSE: Most fish sampled were adults - 171 specimens, with only nine juveniles captured. Of the total of adults, 98% (168 specimens) were sampled in the main channel (sites 4, 26, 27) and 2% (3 specimens) in the tributaries (sites 10, 11, 25). Regarding juveniles, 78% (7 specimens) occurred in the main channel (sites 26, 27) and 22% (2 specimens) in the tributaries (sites 18, 25). We included this information in the results.

6. Conclusions. Acceptable.

Reviewer 2 Report

The work of Lucileine et al. evaluates the spatial distribution and abundance of Steindachneridion melanodermatum in Iguacu River, an endemic and endangered species.

The title is relevant and the abstract is concise.

The introduction is well written and sufficient citations are provided.

2. Materials and Methods 
2.1. Study area

Figure 1 was obtained with a software, please state the software used.

Have you recorded the male/female ratio? If so, please present.

Line 246-249: The percentage given has to be clarified. Provide detailed explanations as in how these percentages were obtained.

The discussion section is relevant and well written

Conclusions are somewhat "hollow" please be concise in the conclusions drawn from your work.

Author Response

Dear Reviewer,

We agree with the comments and suggestions, and we did a review of the manuscript (Water – 1253080) according to requests. The corrections and adjustments are marked up using the "Track Changes" function. The comments and our responses to them are below.

We thank the comments and suggestions to the manuscript, which contributed to its improvement.

Yours sincerely,

Lucileine de Assumpção
Universidade Estadual do Oeste do Paraná

----------------------------------------------------------------------------------------

Reviewer #2 - Comments and Suggestions for Authors

The work of Lucileine et al. evaluates the spatial distribution and abundance of Steindachneridion melanodermatum in Iguacu River, an endemic and endangered species.

The title is relevant and the abstract is concise.

The introduction is well written and sufficient citations are provided.

  1. Materials and Methods

2.1. Study area

Figure 1 was obtained with a software, please state the software used.

RESPONSE: We included the software used – QGIS in the legend of figure 1.

Have you recorded the male/female ratio? If so, please present.

RESPONSE: Yes, we have recorded the sex ratio. However, population and reproductive biology parameters, including sex ratio, are in the paper published – Assumpção et al. (2021). Thus, we don't include these results in the present manuscript.

Assumpção, L.; Fávaro, L.F.; Makrakis, S.; Silva, P.S.; Pini, S.F.R.; Kashiwaqui, E.A.L.; Makrakis, M.C. Population structure and reproduction of Steindachneridion melanodermatum (Siluriformes: Pimelodidae), a large endemic catfish to Neotropical ecoregion. Mar. Freshw. Res. 2021. https://doi.org/10.1071/MF19373

Line 246-249: The percentage given has to be clarified. Provide detailed explanations as in how these percentages were obtained.

RESPONSE: We clarified it.

The discussion section is relevant and well written

Conclusions are somewhat "hollow" please be concise in the conclusions drawn from your work.

RESPONSE: We improved the conclusion.

Round 2

Reviewer 1 Report

Dear authors. My comments are taken into account.